# Anodal Capture for Multisite Pacing with a Quadripolar Left Ventricular Lead: A Feasibility Study

**DOI:** 10.3390/jcm10245886

**Published:** 2021-12-15

**Authors:** Alexandre Bodin, Arnaud Bisson, Clémentine Andre, Dominique Babuty, Nicolas Clementy

**Affiliations:** Service de Cardiologie, Centre Hospitalier Universitaire Trousseau et EA7505, Faculté de Médecine, Université François Rabelais, 37000 Tours, France; arnaud.bisson37@gmail.com (A.B.); clementine2andre@gmail.com (C.A.); d.babuty@chu-tours.fr (D.B.); N.CLEMENTY@chu-tours.fr (N.C.)

**Keywords:** cardiac resynchronization therapy, multipoint pacing, multisite pacing, anodal capture

## Abstract

Background: Up to 40% of patients are CRT non-responders. Multisite pacing, using a unique quadripolar lead, also called multipoint/multipole pacing (MPP), is a potential alternative. We sought to determine the feasibility of intentional anodal capture using a single LV quadripolar lead, to reproduce MPP without the need of a specific algorithm (so-called “pseudo MPP”). Methods: Consecutive patients implanted with a commercially available CRT device and a quadripolar LV lead in our department were prospectively included. The electric charge (Q, in Coulomb) of RV and LV pacing spikes were calculated for all available LV pacing configurations at the threshold. The best MPP was defined as the configuration with the lowest consumption (Q_RV_ + Q_best LV1_ + Q_best LV2_). The best “pseudo MPP” (Q_RV_ + Q_LV1–LV2 with anodal capture_) and best BVp (Q_RV_ + Q_best LV_) were also calculated. A theoretical longevity was estimated for each configuration at the threshold without a safety margin. Results: A total of 235 configurations were tested in 15 consecutive patients. “Pseudo-MPP” was feasible in 80% of patients with 3.1 ± 2.6 vectors available per-patient and LV_proximal_-LV_distal_ (most distant electrodes) vectors were available in 47% of patients. Each MPP pacing spike electrical charge was comparable to “pseudo-MPP” (18,428 ± 6863 µC and 20,528 ± 5509 µC, respectively, *p* = 0.15). Theoretical longevity was 6.2 years for MPP, 5.6 years for “pseudo-MPP” and 13.7 years for BVp. Conclusions: “Pseudo MPP” using intentional anodal capture with a quadripolar left ventricular lead, mimicking conventional multisite pacing, is feasible in most of CRT patients, with comparable energy consumption. Further studies on their potential clinical impact are needed.

## 1. Introduction

Cardiac resynchronization therapy (CRT) in heart failure showed a diminution of mortality and morbidity [1,2,3,4,5,6]. However, up to 40% of patients are non-responders to CRT [2,7,8,9].

The initial concept of LV multisite pacing (MSp) emerged in this context and showed that pacing using 2 LV leads was associated with improved LV reverse remodeling, as compared with standard biventricular pacing (BVp) [10], especially in cases of a posterolateral scar [11]. MSp through a single quadripolar lead (using 2 pacing cathodes out of 4 electrodes), also called multipoint/multipole pacing (MPP), is a safer and easier technique [12,13,14], which showed an improvement in hemodynamics and the functional status but remains debated and still needs more morbimortality evidence, especially through programming with large anatomical separation [15,16,17,18,19,20,21,22,23]. Other limitations include a faster battery drain [24] and the need for a dedicated mode that is programmable within the device software.

Anodal capture results, during bipolar pacing, from a high density current from the cathode allows for the capturing of the myocardium near the anode. Depolarization arises from both the anode and the cathode of the used electrical bipole. Intentional anodal capture during bipolar pacing by the LV quadripolar lead, so-called “pseudo-MPP” (Figure 1), may have an acute hemodynamic benefit equal to conventional MPP [25,26]. 

We sought to determine the feasibility of the intentional anodal capture using a single LV quadripolar lead, to reproduce MPP without the need for a specific algorithm.

## 2. Methods

Consecutive patients implanted with a CRT device and a quadripolar LV lead in our department were prospectively included. Commercially available MPP-capable CRT device from Abbott^®^ (Chicago, IL, USA), Boston Scientific^®^ (Marlborough, MA, USA) and Medtronic^®^ (Dublin, Ireland) were used.

Enrolled patients provided oral informed consent and data were collected in accordance with institutional guidelines on ethics.

Electrical and ECG tests were performed after implantation. Threshold and impedance were manually measured in each programmable LV configuration. The pacing duration was 0.4 milliseconds. “Pseudo-MPP” stimulation was identified by a QRS morphology resulting from the fusion of the unipolar-cathode paced QRS complex and the unipolar-anode paced QRS complex. Its threshold was identified using an LV true bipolar vector decremental threshold test (Figure 2). Vectors with phrenic nerve stimulation (PNS) were not considered. When no ECG difference was observed, either because of absence of “pseudo-MPP” or due to a significant difference of morphology between cathodal and anodal pacing, the vector was not considered.

## 3. Theoretical Battery Drain at the Threshold

Using the data of the threshold tests, the electric charge (Q, in Coulomb) of the RV and LV pacing spikes were calculated for all available LV-pacing configurations at the threshold at 0.4 ms (without any safety margin). The first LV vector for MPP was named LV1, the second was named LV2. The best MPP was defined as the configuration with the lowest consumption (Q_RV_ + Q_best LV1_ + Q_best LV2_). The best “pseudo MPP” (Q_RV_ + Q_LV1–LV2 with anodal capture_) and best BVp (Q_RV_ + Q_best LV_) were also calculated. A theoretical longevity was estimated for best BVp, best MPP and best “pseudo-MPP” configurations in order to compare the battery drain. We considered permanent (100%) RV and LV pacing at 60 beats per minute in the VVI mode without atrial pacing, without any electrical shock, at the threshold without a safety margin, with a pacing duration of 0.4 ms, and a usable capacity of 1000 mAh.

After all tests were performed, the device was programmed according to the standard practice.

## 4. Statistical Analyses

All statistical analyses were performed using JMP^®^ 9.0.1. software (SAS Institute Inc., Cary, NC, USA). Qualitative variables are described using counts and percentages and continuous quantitative variables as mean ± standard deviation. Comparisons between MPP, pseudo-MPP and BVp were performed using non-parametric tests. Statistical significance was assumed at *p* < 0.05. 

## 5. Results

A total of 235 configurations were tested in 15 consecutive patients (5 with an Abbott^®^, 5 with a Boston^®^ and 5 with a Medtronic^®^ device).

The baseline characteristics are described in Table 1. Patients were 72 ± 11 years old, 67% were male, 40% had ischemic heart disease, 60% had a left bundle branch block morphology, the mean intrinsic QRS duration was 137 ± 27 ms and the mean left ventricular ejection fraction (LVEF) was 28 ± 6%.

Pacing characteristics are described in Table 2. The final LV lead location was lateral in 47% of patients. “Pseudo-MPP” was feasible in 80% of patients, with 3.1 ± 2.6 vectors available per patient (35% of all LV bipolar vectors, i.e., 47/135). “Pseudo-MPP” with LV_proximal_-LV_distal_ (most distant electrodes) vectors was available in 47% of patients (45% of the vectors, i.e., 9/20). The mean “pseudo-MPP” threshold was 5.2 ± 0.9 V. Only two vectors with PNS during the “pseudo-MPP” test were excluded.

At the threshold without a safety margin, at 0.4 ms, each MPP pacing spike electrical charge was comparable to “pseudo-MPP” (18,428 ± 6863 µC and 20,528 ± 5509 µC, respectively, *p* = 0.15). The theoretical longevity was 6.2 years for MPP, 5.6 years for “pseudo-MPP” and 13.7 years for BVp (Table 3).

The mean RV electrical charge was 3788 ± 1300 µC, representing 20.6% for MPP, 18.5% for “pseudo-MPP” and 45.3% for BVp, of the total biventricular electrical charge.

## 6. Discussion

In this pilot study, the main results were: (1) “pseudo-MPP” is feasible in 80% of patients, in almost half of the patients when using the most distant electrodes; (2) “pseudo-MPP” and MPP energy consumptions are comparable.

Dell’Era et al. enrolled 30 CRT patients and tested, during the implantation procedure, the anodal capture on quadripolar leads from the three main manufacturers, using a Medtronic pacing-sensing analyzer [27].

Conversely, we evaluated the feasibility of “pseudo-MPP” after implantation using the implanted CRT devices algorithms. They found an anodal capture in 93% of the patients, with a maximum pacing impulsion of 10 V at 0.5 ms, a much higher value than the one programmable in “real life”, explaining the lower 80% rate in our study. The mean anodal capture threshold was 3.9 ± 2.4 V at 0.5 ms (5.2 ± 0.9 V at 0.4 ms in our study) and the mean cathodal capture threshold was 1.9 ± 1.6 V at 0.5 ms (2.2 ± 0.8 V at 0.4 ms in our study), which was found to be comparable in both studies.

“Pseudo MPP” has several advantages. It allows for MPP in MPP incompatible devices. Anodal capture is also associated with absence of intraventricular delay (0 ms), which could be useful in devices with a minimum programmable delay (5 ms for Abbott and 10 ms for Medtronic for example), and may achieve a better resynchronization [21].

We do not provide data on any hemodynamics improvement. Morishima et al. showed an acute improvement on dP/dT max during “pseudo-MPP” comparable to MPP [25]. 

The benefits of MPP and more so of “pseudo-MPP” remain unclear, and acute hemodynamics improvement does not predict the long-term response and/or clinical outcomes.

## 7. Limitations

Anodal capture is time consuming, necessitating a 12-lead electrocardiogram and a manual threshold test, which may be difficult to perform routinely. Very limited changes in the QRS morphology during “pseudo-MPP” threshold tests were observed in a few cases, with a difficult determination of the true “pseudo-MPP” threshold and probably limited effects on the depolarization wavefront and resynchronization.

“Pseudo-MPP” and MPP are associated with rapid battery depletion as compared with conventional BVp. Indeed, the battery drain was about twice the difference versus BVp. A prospective study of the selected patients would be interesting in order to detect potential clinical impact. However, high battery drain would be associated with frequent generator changes and associated risks (infectious for example). In our opinion, only non-responder patients to CRT could ethically be included by comparing the best BVp versus MPP and “pseudo-MPP”. 

## 8. Conclusions

“Pseudo MPP”, using intentional anodal capture with a quadripolar left ventricular lead, mimicking conventional multisite pacing, is feasible in most CRT patients, with a comparable energy consumption. 

Further studies on the potential clinical impact are needed.

## Figures and Tables

**Figure 1 jcm-10-05886-f001:**
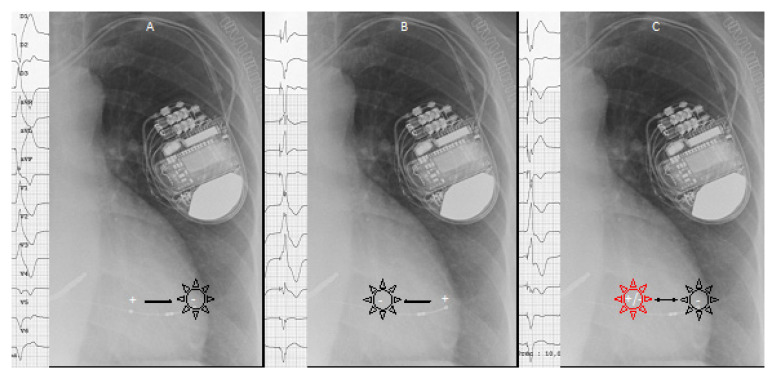
Intentional anodal capture during bipolar pacing by the LV quadripolar lead, so-called “pseudo-MPP”. During conventional bipolar pacing, depolarization wave front arise from the cathode (−) (**A**,**B**). When a high-density current is applied, anodal capture may be achieved, and depolarization wavefront arises from both the anode (−) and the cathode (+) of electrical bipole. The resulting QRS complex (left panels) is a fusion between anodal and cathodal pacing (**C**).

**Figure 2 jcm-10-05886-f002:**
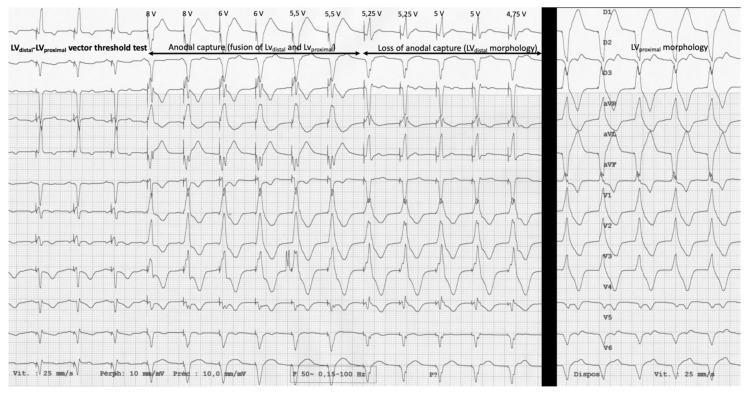
“Pseudo MPP” threshold measurement. Example with LV_distal_ − LV_proximal_ vector threshold test with an impulsion duration of 0.4 ms. An abrupt change in QRS morphology can be identified between 5.5 V and 5.25 V, 5.5 V being considered as the “pseudo-MPP” threshold. “Pseudo-MPP” QRS morphology results from the fusion of mono-LVp_distal_ and mono-LVp_proximal_ QRS morphologies.

**Table 1 jcm-10-05886-t001:** Baseline characteristics.

Patients, *n*	15
Age, years	72 ± 11
Male sex, *n* (%)	10 (67%)
Ischemic heart disease, *n* (%)	6 (40%)
Hypertension, *n* (%)	8 (53%)
Diabetes mellitus, *n* (%)	4 (27%)
Sinus rythm, *n* (%)	13 (87%)
LBBB, *n* (%)	9 (60%)
QRS duration (ms)	137 ± 27
LVEF (%)	28 ± 6
LVEDD (mm)	58 ± 6
*Device*	
Abbott, *n*	5
Boston, *n*	5
Medtronic, *n*	5
*Quadripolar lead*	
Abbott’s Quartet, *n* (%)	3 (20%)
Boston’s Acuity, *n* (%)	4 (27%)
Medtronic’s Attain, *n* (%)	8 (53%)

LBBB: Left bundle branch block, LVEF: left ventricular ejection fraction, LVEDD: Left ventricular end-diastolic diameter.

**Table 2 jcm-10-05886-t002:** Pacing characteristics.

*Final LV lead location*	
Anterior, *n* (%)	1 (6%)
Anterior-lateral, *n* (%)	4 (27%)
Lateral, *n* (%)	7 (47%)
Posterior-lateral, *n* (%)	2 (14%)
Posterior, *n* (%)	1 (6%)
Mean LV threshold (V)	2.2 ± 0.8
Best LV threshold (V)	0.9
Mean LV vector impedance (Ohm)	731 ± 309
Available “pseudo MPP” vectors, *n*	3.1 ± 2.6
Mean “pseudo MPP” threshold (V)	5.2 ± 0.9
Mean LV1 threshold when “pseudo MPP” available (V)	2 ± 0.6
Mean LV2 Threshold when “pseudo MPP” available (V)	1.6 ± 0.5

LV: left ventricular, LV1: first LV vector, LV2: second LV vector, MPP: multipoint/multipole pacing.

**Table 3 jcm-10-05886-t003:** Mean Electrical charge (µC) and theoretical longevity (years) at the threshold without safety margin.

	Best MPP	Best “Pseudo MPP”	Best BVp
Electrical charge (µC)	18,428 ± 6863	20,528 ± 5509 **∫**	8357 ± 2307 **‡**
Longevity (years)	6.2	5.6	13.7

MPP: multipoint/multipole pacing, BVp: biventricular pacing. **∫**
*p* = 0.15 best “pseudo MPP” versus best MPP, **‡**
*p* = 0.0005 best BVp versus best MPP and best “pseudo MPP”.

## Data Availability

Data is contained within the article.

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
