# Peer review of "Anodal Capture for Multisite Pacing with a Quadripolar Left Ventricular Lead: A Feasibility Study"

_jcm, 2021, doi:10.3390/jcm10245886_

Round 1

Reviewer 1 Report

The report by Alexandre B. presents the feasibility of intentional anodal capture using an LV quadripolar lead. Multipoint pacing has a potential role for more favorable outcomes compared with the conventional LV pacing methods, and the present new technique “pseudo-multipoint pacing” may also have the same potentials. Amazingly, pseudo-MMP could be achieved in 80% of patients, and the energy consumptions of the devices are the same as the conventional MMP methods. These results suggest that pseudo-MMP might play a role in the improvement of the CRT responder rate especially in patients with large scar areas. However, there are substantial issues raised in the following comments that need to be addressed.

  • In my most concern, the pseudo-MMP may induce phrenic nerve stimulation (PNS) especially in cases of using the most distant electrodes. Owing it is well known that the distance of electrodes is the key for avoiding PNS, the pseudo-MMP pacing using the most distant electrodes must capture the phrenic nerve. The authors described that the vectors with PNS were not considered; the reviewer believes the PNS is the key barrier for applying the pseudo-MMP to clinical practice, however. Therefore, the authors should clarify the incidence rate of PNS during pseudo-MMP.

  • The reviewer speculates that pseudo-MPP threshold or suitable pacing electrodes may change with LV reverse remodeling. The authors emphasized the benefits of the methods without a specific algorithm, how detect those changes in order to generalization to CRT patients?

  • The differences in battery longevity between best BVp and MMP including pseudo-MMP stand out (about twice the difference!). The authors should discuss the comparison of favorable effects for hemodynamics of MMP and the frequent generator changes with an infectious potential.

Reviewer 2 Report

I had the pleasure to review the manuscript “Anodal capture for multisite pacing with a quadripolar left ventricular lead; a feasibility study” by BODIN Alexandre et al. The authors show a very interesting way to simulate multipoint pacing in systems that are not capable to use this new technology, just by increasing stimulation output and therefore enabling anodal capture. They confirm that battery drain of “pseudo MPP” is similar to actual MPP. In my opinion, this study is very well performed. I just have the following minor comments:

  • Please add into the results that Pseudo MPP could be achieved in 93% with output of 10V/0.5ms.
  • Results: Was this QRS the intrinsic QRS duration?
  • Did you actually enable “pseudo MPP” in a proportion of patients tested?
  • What was the rate of responders to CRT therapy in your cohort?
  • What was the age of devices used? Would a proportion of devices have been able to perform classical MPP?
  • Had “pseudo MPP” an advantage of QRS shortening in your cohort? Please add the parameter “mean QRS duration” for native QRS (if not pacing-dependent), QRS (classically stimulated) and QRS (Pseudo MPP).

Round 2

Reviewer 1 Report

The manuscript has been much improved and is in nice condition now.